# Exploring Genomic Biomarkers for Pembrolizumab Response: A Real-World Approach and Patient Similarity Network Analysis Reveal DNA Response and Repair Gene Mutations as a Signature

**DOI:** 10.3390/cancers16233955

**Published:** 2024-11-26

**Authors:** Marco Filetti, Mario Occhipinti, Alessio Cirillo, Fabio Scirocchi, Alessio Ugolini, Raffaele Giusti, Pasquale Lombardi, Gennaro Daniele, Andrea Botticelli, Giuseppe Lo Russo, Filippo De Braud, Paolo Marchetti, Marianna Nuti, Elisabetta Ferretti, Lorenzo Farina, Aurelia Rughetti, Manuela Petti

**Affiliations:** 1Phase 1 Unit, Fondazione Policlinico Universitario Agostino Gemelli, IRCCS, 00168 Rome, Italy; pasquale.lombardi89@gmail.com (P.L.); gennaro.daniele@policlinicogemelli.it (G.D.); 2Department of Experimental Medicine, Sapienza University of Rome, 00161 Rome, Italy; mario.occhipinti@uniroma1.it (M.O.); alessio.cirillo@uniroma1.it (A.C.); fabio.scirocchi@uniroma1.it (F.S.); alessio.ugolini@uniroma1.it (A.U.); marianna.nuti@uniroma1.it (M.N.); elisabetta.ferretti@uniroma1.it (E.F.); aurelia.rughetti@uniroma1.it (A.R.); 3Thoracic Oncology Unit, Fondazione IRCCS Istituto Nazionale dei Tumori, 20133 Milan, Italy; giuseppe.lorusso@istitutotumori.mi.it; 4Department of Radiology, Oncological and Anatomo-Pathological Science, Sapienza University of Rome, 00161 Rome, Italy; andrea.botticelli@uniroma1.it; 5Department of Onco-Hematology, Gene and Cell Therapy, Bambino Gesù Children’s Hospital–IRCCS, 00165 Rome, Italy; 6Department of Immunology, H. Lee Moffitt Cancer Center, Tampa, FL 33612, USA; 7Department of Medical Oncology, St. Andrea Hospital, 00189 Rome, Italy; raffaelegiusti@yahoo.it; 8Oncology and Hemato-Oncology Department, University of Milan, 20133 Milan, Italy; filippo.debraud@istitutotumori.mi.it; 9Istituto Dermopatico dell’Immacolata IDI-IRCCS, 00167 Rome, Italy; paolo.marchetti@uniroma1.it; 10Department of Computer, Control, and Management Engineering, Sapienza University of Rome, 00161 Rome, Italy; lorenzo.farina@uniroma1.it (L.F.); manuela.petti@uniroma1.it (M.P.)

**Keywords:** immunotherapy, DNA damage response and repair, non-small cell lung cancer, never-smoker, network analysis

## Abstract

This study investigates the molecular characterization of never-smoking patients (NS-pts) with high tumor mutation burden (H-TMB) in advanced non-small cell lung cancer (aNSCLC) and their response to immune checkpoint inhibitors (IO). Clinical data from 142 aNSCLC patients with PD-L1 ≥ 50% treated with pembrolizumab were analyzed. Next-generation sequencing identified mutations across 11 main pathways, revealing that NS-pts with H-TMB were enriched in β-catenin/Wnt and DDR pathway mutations. Validation using data from the POPLAR and OAK trials confirmed DDR pathway mutations were significantly associated with improved outcomes in NS/H-TMB patients. Patient similarity network analysis further showed that NS-pts with DDR mutations had better overall survival. The findings suggest that DDR mutations contribute to H-TMB in NS-pts and may identify a subgroup of patients who could benefit from IO therapy, offering potential for improved prognosis and targeted treatment strategies.

## 1. Introduction

Pembrolizumab has emerged as the standard of care for stage IV, non-oncogene-addicted advanced non-small cell lung cancer (aNSCLC) patients with programmed death-ligand 1 (PD-L1) expression ≥ 50% [1]. However, the response to pembrolizumab treatment varies significantly among patients, underscoring the need for additional biomarkers to aid in patient selection. One potential biomarker is tumor mutation burden (TMB), which quantifies the total number of nonsynonymous mutations per sequenced coding area of a tumor genome and has shown predictive value for immunotherapy (IO) efficacy across various tumor types [2]. Recent work by Ricciuti et al. demonstrated a correlation between increasing TMB levels and immune cell infiltration, as well as an inflammatory T-cell-mediated response. In aNSCLC, this enhanced immune activity results in increased sensitivity to PD-1/PD-L1 inhibitors in both subgroups with PD-L1 expression above and below 50% [3]. It is worth noting that TMB is influenced by multiple factors, including smoking status. Smoking-related damage results in high TMB (H-TMB) in aNSCLC patients who smoke (S-pts), while never-smoking patients (NS-pts) generally exhibit low TMB (L-TMB) and are considered less responsive to IO [4]. Consequently, there is an urgent need to identify biomarkers that can improve the stratification of aNSCLC patients, particularly NS-pts, and thereby enhance the efficacy of IO-based treatments.

In addition to TMB, several mutations in signaling pathways have been proposed as potential predictors of IO response or resistance in aNSCLC. For instance, alterations in cell cycle regulators, such as MDM2/MDM4 amplifications, have been suggested as markers of IO resistance [5,6]. On the other hand, mutations in DNA damage response and repair (DDR) genes are associated with increased genomic instability and H-TMB in cancer, which may enhance immunogenicity through a higher tumor-specific neoantigen load [7,8,9,10,11,12]. Deleterious DDR mutations are frequent in aNSCLC and have been linked to improved clinical outcomes in patients treated with PD-L1 blockade [13]. Despite these promising findings, further investigation is needed to explore the potential role of signaling pathway mutations, in combination with TMB and clinical features, for predicting response outcomes in real-world populations.

Therefore, the aim of this study was to identify potential genomic biomarkers for pembrolizumab response by evaluating the impact of signaling pathway mutations and TMB, in conjunction with clinical characteristics, on patient outcomes in a real-world setting. Furthermore, we sought to validate our findings in a larger cohort of aNSCLC patients by leveraging publicly available databases and constructing a patient similarity network. The results of this study have the potential to inform clinical decision-making and improve outcomes for aNSCLC patients undergoing pembrolizumab treatment.

## 2. Materials and Methods

### 2.1. Real-World Cohort

This retrospective study was conducted at Sapienza University in Rome, Italy. The internal review board and local ethics committee approved the study protocol (Protocol number: 297 SA_2019). Patient data were collected in conformance with the principles of the Declaration of Helsinki, Good Clinical Practice guidelines, and local ethical rules. All patients provided written informed consent for use of their clinical data (at any point in their medical history) for research.

The collected variables included: age, Eastern Cooperative Oncology Group performance status score (ECOG PS), gender, ethnicity, smoking history, diagnosis, histology, tumor burden and metastatic sites, comorbidities, and the start and end dates of treatment. Data from 142 consecutive aNSCLC patients were collected retrospectively between January 2017 and March 2021. The following criteria had to be met for inclusion in the study:Cytological or histological diagnosis of aNSCLC (stage IIIB to IV)Receipt of at least one cycle of first-line pembrolizumabECOG PS score of 0–2Availability of tumor tissue for next generation sequencing (NGS) analysis

Treatment was continued until the occurrence of disease progression, unacceptable toxicity, withdrawal of consent, or death. Treatment beyond progressive disease was permitted in cases demonstrating clinical benefit. Radiological assessments consisted of a total-body computed tomography scan, which was performed at baseline, at variable time intervals based on local clinical practice guidelines, and on clinical suspicion of progressive disease (PD). Tumor response was assessed according to Response Evaluation Criteria in Solid Tumors (RECIST) v.1.1, and was defined as complete response (CR), partial response (PR), stable disease (SD), and progressive disease (PD). The objective response rate (ORR) was defined as the sum of CR and PR, while the disease control rate was defined as the sum of CR, PR, and SD.

### 2.2. POPLAR and OAK Population

Freely publicly available data from blood-based NGS (bNGS) of the biomarker-evaluable population (BEP) of the phase II POPLAR (NCT01903993) (N = 211) and phase III OAK (NCT02008227) (N = 642) randomized trials were used for external validation [14,15,16] (N total = 853). The POPLAR BEP included 15 patients with an epidermal growth factor receptor (EGFR) mutation or anaplastic lymphoma kinase (ALK) rearrangement and 196 without known alterations. The OAK BEP included 59 patients with an EGFR mutation or ALK rearrangement, and 583 without known alterations. Both the POPLAR and OAK studies were performed in full accordance with the guidelines for Good Clinical Practice and the Declaration of Helsinki, and all patients had provided written informed consent. Protocol approvals were obtained from independent ethics committees of each participating site for both studies. Similarities in the design of the POPLAR and OAK trials justified pooling of the data. Both trials included patients with measurable previously treated aNSCLC, unselected for PD-L1 status, and randomly assigned to receive either atezolizumab or docetaxel; both used the same stratification factors and schedule of assessments, and crossover was not permitted. In both trials, patients were randomized to intravenous atezolizumab or docetaxel arms in a 1:1 ratio. The primary endpoint was overall survival (OS); progression-free survival (PFS) was one of the secondary endpoints. Detailed descriptions of the eligibility criteria and recruitment methods for both trials have been published previously [15,16,17]. The variables considered were: ECOG PS, gender, smoking history, histology, tumor burden, and the start and end dates of treatment. Each study team reviewed all axial computed tomographic images according to RECIST v.1.1.

### 2.3. Targeted Next-Generation Sequencing and Oncogenic Pathway Classification

For the real-world population, NGS analysis was performed on available pretreatment tumor tissue using the FoundationOne^®^CDx assay (Foundation Medicine, Cambridge, MA, USA).

Alterations reported in the Catalogue of Somatic Mutations in Cancer (COSMIC) and ClinVar databases [18,19] were identified in both populations (real-world and POPLAR/OAK); the alterations reported as pathogenic by the COSMIC and ClinVar were classified as deleterious. A total of 11 signaling pathways with frequent genetic alterations were evaluated, starting with key cancer genes explored in previous TCGA publications [20]. The following pathways were analyzed: (1) cell cycle, (2) Hippo, (3) Myc, (4) Notch, (5) oxidative stress/Nrf2, (6) PI3K, (7) RTK/RAS/MAP, (8) TGF-β, (9) p53, (10) β-catenin/Wnt, and (11) DDR. They were used to redefine the mutational profile of patients in terms of mutated pathways: each pathway was labeled “mutated” when the patient manifested at least one mutation in that pathway. The baseline molecular characteristics of the real-world cohort and all the analyzed genes and pathway member genes considered for the real-world and POPLAR/OAK populations have been listed in Appendix A [13,21,22,23,24,25,26,27,28,29].

### 2.4. PD-L1 Testing and (Blood) TMB Assessment

The same PD-L1 testing and TMB assessments methods were used in the real-world and POPLAR and OAK populations. PD-L1 expression was reported as the percentage of tumor cells with positive membranous staining in a slide containing at least 100 tumor-viable cells. PD-L1 expression was determined by immunohistochemistry using the Dako PD-L1 22C3 pharmDx assay (Dako, Glostrup, Denmark). TMB, defined as the number of somatic, coding, base substitution, and insertion/deletion (indel) mutations per megabase (Mb) of genome examined, was calculated using the NGS FoundationOne^®^CDx assay. The blood TMB (bTMB) assay uses the same hybridization-capture methodology as the United States Food and Drug Administration-approved FoundationOne CDx NGS assay and targets 1.1 Mb of genomic coding sequences [30]. We applied the same TMB cut-off (>10 Mut/Mb) used in previous studies to define high TMB, as validated by earlier research [31].

### 2.5. Clinical Outcomes

PFS was defined as the time from the start of IO therapy to disease progression or death, whichever occurred first. Patients who were alive without disease progression were censored on the last date of adequate disease assessment. OS was defined as the time from the start of IO therapy to death. Patients who were still alive were censored at the date of last contact.

### 2.6. Statistical Analysis

Descriptive statistics were used to characterize the patient cohorts, and the Kaplan-Meier method and log-rank test were used for survival analysis. Between-group differences in clinical characteristics were analyzed using the two-tailed Wilcoxon rank sum test and the Kruskal–Wallis test. Enrichment analysis (hypergeometric test) was performed to characterize patients’ subgroups in terms of mutated pathways. The Benjamini-Hochberg method was applied to control for the false discovery rate (FDR). Statistical analyses were also performed with a focus on the sub-cohorts identified based on TMB levels (H-TMB and L-TMB) and smoking status (S-pts and NS-pts).

### 2.7. Patient Similarity Network Analysis

To further investigate the patients’ molecular characterization and the presence of a relationship between signaling pathways, TMB levels, and smoking history in the real-world and POPLAR/OAK populations, patient similarity networks were constructed for both S- and NS-pts cohorts. In a patient similarity network, the nodes correspond to the patients and the weighted links between nodes indicate how similar the patients are in terms of specific properties [32]. In this study, we evaluated the similarity between patients based on their mutational profiles. In particular, the Euclidean distance was first computed between the mutated pathway profiles of each pair of patients, and then a scaled exponential similarity kernel was used to determine the weight of the network links [33]. Once the network was obtained, community detection (Louvain method) [34,35] was performed to discover subgroups of patients more like each other (network community). The identified communities were finally characterized in terms of clinical characteristics and response outcomes.

## 3. Results

### 3.1. Non-Smoking Patients with H-TMB in a Real-World Cohort Displayed a Novel Genomic Profile Characterized by Mutations in the DDR Pathway, and a Better Outcome to Pembrolizumab Treatment

Overall, 142 patients with aNSCLC were retrospectively enrolled in our real-world cohort between January 2017 and March 2021; the patients’ clinical characteristics are summarized in Table 1.

With a median follow up of 17.05 months, median OS was 18.3 months and median PFS was 12.2 months. A total of 88 (62%) male and 54 (38%) female patients were enrolled; the average age was 63.35 (SD +/− 10.48) years. Most patients were smokers or former smokers (S-pts, N = 111, 78%), while 31 patients had no history of tobacco use (NS-pts, 22%). Most tumors were adenocarcinomas (N = 123, 87%); 19 patients (13%) had squamous histology. The median TMB of the total population was 7.57 Mut/Mb. S-pts had a higher median TMB than NS-pts (8 vs. 4 Mut/Mb, respectively) (Figure 1). Interestingly, among the NS-pts, 11 showed a TMB of higher than 10 Mut/Mb (median TMB: 16.39 Mut/Mb).

A review of all mutations detected by NGS population analysis in the COSMIC and ClinVar databases enabled the identification of mutations reported as pathogenic. Such pathogenic mutations were subsequently classified into the 11 signaling pathways previously described. Table 2 shows the number of S- and NS- patients characterized by at least one mutation in each of the pathways.

Enrichment analysis was then performed to assess whether specific pathways accumulated a higher frequency of mutations depending on patient smoking history (S-pts vs. NS-pts) or the TMB cut-off (H-TMB vs. L-TMB).

On performing enrichment analysis considering the TMB level, the H-TMB group (53 pts, 37%) had statistically significant enrichments in cell cycle and Notch pathway alterations (FDR < 0.05). On the contrary, no enrichment was identified in the L-TMB cohort (Appendix A). The smoking history, independent of the TMB values, did not appear to be associated with alterations of specific pathways; no significant enrichment of mutations was found in any pathway when comparing S-pts with NS-pts (Appendix A).

As previously performed for the general population, enrichment analysis was repeated after stratifying the two cohorts of S-pts and NS-pts based on the level of TMB (H-TMB and L-TMB). Two statistically significant enrichments were identified in the NS/H-TMB cohort. In particular, the highlighted enrichments depended on the β-Catenin/Wnt and the DDR pathways (FDR < 0.05). Eight of eleven (73%) NS/H-TMB patients had at least one alteration of the DDR pathway (Appendix A). These results indicated the presence of distinct molecular profiles (depending on the TMB values) in the entire patient population (H-TMB vs. L-TMB patients) and within the specific subgroup of NS/H-TMB patients.

We then investigated whether these distinct molecular profiles in S-pts and NS-pts were associated with different clinical outcomes. Indeed, among H-TMB patients, the NS/H-TMB subgroup, which was enriched by DDR and β-catenin mutations, exhibited a remarkable median OS of 27.95 months and an impressive overall response rate (ORR) of 100% (11/11 patients) (Figure 2A,B). On the other hand, among the L-TMB patients, the NS/L-TMB patient displayed a worse clinical outcome than the S/L-TMB patients (Appendix A). These results indicated that highly effective response to pembrolizumab treatment is observed in NS/H-TMB patients enriched by the DDR and β-catenin mutations.

### 3.2. DDR Pathway Mutations as a Molecular Feature of NS/H-TMB Patients in POPLAR/OAK Populations

To validate the molecular features observed in the highly responsive NS/H-TMB patients of the real-world cohort, we used the bNGS data from the BEP of the phase II POPLAR (NCT01903993) and III OAK (NCT02008227) randomized trials. Although tumor and blood profiles may show some differences, they are highly correlated and can be compared, with the understanding that the mutation frequencies in blood are generally lower than those observed in tumor samples.

Clinical and molecular data from a total of 853 patients were analyzed. Among them, 713 (84%) patients were current or former smokers, while 140 (16%) were never-smokers. The cohorts of S-pts and NS-pts had a median TMB of 9 and 4 Mut/Mb, respectively (Appendix A). All the pathogenic mutations in the 11 pathways for the POPLAR/OAK cohort were classified and enrichment analyses were performed as previously described.

Of the NS-pts cohort, 18 patients were found to have H-TMB (>10 Mut/Mb): this subgroup was employed to validate the molecular profile of DDR and β-catenin mutation pathways. While no significant enrichment was observed for the mutation in the β-catenin/Wnt pathway (five of eighteen patients, *p*-value = 0.6129), mutations in the DDR pathway were detected in seventeen of eighteen patients from the NS/H-TMB subgroup, and six of them had more than one mutation (2–8 mutations). These results validated the significant enrichment for mutations in the DDR pathway (*p*-value = 0.0297) in the NS/H-TMB patients.

### 3.3. Network Analysis for Patient Similarity Evaluations

To further investigate the molecular characterization of the NS-pts in our cohort (31 patients) and validate the DDR mutation pathway as a peculiar molecular feature of the NS/H-TMB patients group, we took advantage of a network analysis approach. This computational approach allowed us to build a network based on the molecular similarity of the patients (Figure 3A–C). The network resulted in four communities (C1–4), each characterized by a distinct profile of mutated pathways (Figure 3A,C). The first subgroup (NS C1) comprised eight patients, all with a TMB level above the threshold of 10 Mut/Mb.

These patients showed similarity due to the mutation of several pathways in more than half of them (Figure 3C). These pathways included DDR (six of eight patients), the cell cycle (seven of eight patients), RTK/RAS/MAP (seven of eight patients), p53 (seven of eight patients), Notch (six of eight patients), oxidative stress/Nrf2 (five of eight patients), and β-catenin/Wnt (five of eight patients). The second community (NS C2, four patients) showed a clear pattern of similarity; all patients had at least one mutation in both RTK/RAS/MAP and p53 pathways. The third community (NS C3, seven patients) was characterized by mutations in the cell cycle, RTK/RAS/MAP, and p53 pathways in all patients (100%). The fourth community (NS C4, 12 patients) had a weaker profile of mutated pathways. The highest percentage of patients shared at least one mutation of the RTK/RAS/MAP pathway (eight of twelve patients); the other percentages were below 50% (Figure 3C).

Notably, the NS C1 community showed a significantly higher TMB level than the other communities (*p*-value = 0.0025) (Figure 3B), and most of the patients carried DDR mutations. When the clinical features of these four similarity network communities were analyzed, an interesting trend was observed in OS. Indeed, when focusing on the communities with a clear mutational profile (NS C1, NS C2, NS C3), by using the Kaplan-Mayer method to estimate OS, it becomes apparent that NS C1 and NS C2 communities had a significantly higher survival rate than the NS C3 community (*p* = 0.022).

These findings emphasize the distinctness of these two communities and suggest that they have more favorable prognoses (Appendix A). No significant trend emerged in terms of PFS. Similarity network analysis was also performed in the S-pts population and obtained four communities that shared commonly mutated pathways (Figure 4A–C). The S C2 community (14 patients) had mutations in the RTK/RAS/MAP and p53 pathways. The PI3K pathway was also found to be mutated in the S C3 community (eight patients, 100% mutated), while the S C4 community (77 patients) did not show a clear profile of mutated pathways; the highest percentages of patients shared mutations in the RTK/RAS/MAP (67.53%; 52/77 pts) and p53 (66.23%; 51/77 pts) pathways (Figure 4A,C). Despite the different molecular profiles, none of the S-pts communities identified were characterized by a significantly higher level of TMB; the threshold of 10 Mut/Mb was also not consistent (i.e., all patients with a TMB level above the threshold, as in NS C1). Indeed, the S C2 community demonstrated lower levels of TMB than the average population (7.57 Mut/Mb) and the S C3 and S C4 communities (*p*-value = 0.0422) (Figure 4B). When the clinical features of the S communities were analyzed, no significant trend emerged in terms of OS and PFS. These results obtained by network patient analysis confirmed that NS habit, H-TMB and DDR mutations clustered together were associated with better clinical response.

Finally, we investigated the community structure of patient similarity networks using the mutational profiles of POPLAR/OAK patients. In the NS-pts cohort, we identified nine network communities (Appendix A). It is worth noting the presence of one community consisting of 57 patients, which exhibited a mutational profile strongly correlated (Spearman correlation; rho = 0.727, *p*-value = 0.0178) with the mutational profile of the real-world NS C1 community (Figure 5). This community was significantly enriched in NS-pts with a TMB level above the threshold of 10 Mut/Mb (hypergeometric test, *p*-value = 0.003). Out of the 18 NS-pts with H-TMB, 14 of them were part of this community. Among the other communities, we also identified one composed of six patients, all of whom had at least one mutation in both the RTK/RAS/MAP and p53 pathways (equivalent to the NS C2 community in the real-world population). Additionally, we want to highlight the presence of a community consisting of eight patients, with only one of them having H-TMB, but all of them having mutations solely in the DDR pathway. No significant trends emerged among the nine communities in terms of TMB, OS, and PFS. On the other hand, the similarity network of POPLAR/OAK smoking patients did not exhibit a community structure.

Therefore, the network analysis we conducted allowed us to identify distinct patient communities based on their molecular features and smoking status. Notably, this analysis provided further confirmation that mutations in the DDR pathway, which are associated with increased OS, were a characteristic molecular trait of patients belonging to the NS C1 community, specifically NS patients with high TMB. This finding highlights the potential clinical significance of this patient cohort. However, our approach also suggests that the presence of DDR mutations alone was not sufficient to fully characterize the NS subtype. In fact, in the larger POPLAR/OAK dataset, we identified a group of eight patients with low TMB (except for one patient) who had mutations only in the DDR pathway.

## 4. Discussion

The present study focused on a real-world cohort of treatment-naive patients with aNSCLC and PD-L1 expression of 50% or higher. We aimed to identify molecular subgroups within this population and investigate their response to IO therapy. This study revealed the existence of a subgroup of non-smoking patients characterized by H-TMB and a distinct molecular profile. Interestingly, unlike non-smoking patients typically characterized by L-TMB and poor IO response, this subgroup of NS/H-TMB patients displayed improved clinical outcomes when treated with first-line pembrolizumab. To the best of our knowledge, this study represents the first real-world report identifying a specific molecular subgroup among treatment-naive patients with aNSCLC and PD-L1 ≥ 50%. These findings were further confirmed in an external cohort using data from the POPLAR/OAK trials using a patient similarity network analysis, which revealed distinct patient communities based on their molecular features and smoking status. Notably, within the non-smoking cohort, a community comprising all patients with high TMB was identified, and this community was enriched with DDR mutations. This result underscores that the community is not only united by a high level of TMB, but also shares molecular characteristics, likely indicating common carcinogenesis processes.

Prior studies have explored the role of DDR mutations as predictors of responses to IO through retrospective and in-silico analyses [3,4,5,6,7,8,9,10,36]. Alterations in genes associated with DDR, homologous recombination repair (HRR), and mismatch repair (MMR) systems have been identified as predictive factors for IO response in lung cancer patients [37]. However, studies have not primarily focused on establishing a relationship between the DDR pathway and IO response. Additionally, these studies typically analyzed DDR alterations in the overall patient population and did not identify specific enrichments in non-smoking patients.

Although the link between lung cancer and smoking is well-established, the mechanisms underlying the development of non-smoking NSCLC are less understood. Importantly, non-smoking NSCLC cases are increasingly common worldwide. Recent data have presented a potential etiological mechanism for EGFR-mutant non-smoking NSCLC [38]. Nevertheless, a deeper understanding of the carcinogenesis and molecular classification of non-smoking NSCLC remains an unmet clinical need. Non-smoking aNSCLC patients usually exhibit driver mutations and L-TMB, rendering IO treatments ineffective or even counterproductive. In this study, we identified a molecular subgroup of non-smoking patients with H-TMB who lacked driver mutations but exhibited widespread genomic instability in the DDR pathway. These alterations could potentially explain the occurrence of carcinogenesis without other risk factors such as smoking or alterations in driver genes like EGFR and ALK. Additionally, these findings suggest the identification of a subgroup of aNSCLC patients prone to genomic instability. The evidence presented in this study was validated using an external cohort of aNSCLC patients and through patient similarity network analysis. Notably, the analysis revealed a distinct community within the non-smoking cohort comprising all patients with H-TMB, characterized by an enrichment of DDR mutations and a homogeneous molecular profile that sets them apart from all other non-smoking patient communities identified. This finding indicates that the community is united not only by H-TMB levels but also by shared molecular characteristics, likely indicating common carcinogenesis processes. The increasing availability of comprehensive genome profiling presents an opportunity to further identify and characterize this molecular subgroup in detail.

However, the current analysis was limited by a small sample size, preventing the determination of specific and recurrent DDR pathway mutations. Additionally, the study was retrospective and conducted at a single center, and analyses were not available to verify potential germline alterations in the DDR pathway. In fact, we cannot exclude that at least some of these patients may carry germline mutations in the DDR pathway, an event that could lead to high genetic instability and predisposition to early-onset cancers.

Despite these limitations, the results hold important biological and clinical relevance. DDR mutations in this NS/H-TMB subgroup should not be solely considered as a proxy for H-TMB, but rather as potential key events in carcinogenesis. Indeed, this subgroup demonstrates a stronger response to immunotherapy than patients with high TMB. These patients may possess underlying genetic instability mechanisms that enhance their responsiveness to immunotherapeutic treatment.

The identification of this molecular cohort could have significant therapeutic implications, particularly with the recent development of various molecules targeting the DDR pathway, such as PARP and ATR inhibitors, either alone or in combination with immunotherapeutic agents in solid tumors [39,40,41].

## 5. Conclusions

In conclusion, the current study identified a molecular subgroup of non-smoking patients with aNSCLC exhibiting H-TMB and harboring DDR pathway mutations. These patients may respond favorably to IO therapy, suggesting that DDR mutations could serve as a potential biomarker for selecting non-smoking patients with non-oncogene-addicted aNSCLC and PD-L1 ≥ 50% for IO treatment. However, these findings need validation through larger, multicenter prospective studies to confirm the data and explore the role of pathogenic DDR mutations in patients eligible for single-agent IO therapy or IO in combination with chemotherapy. If confirmed, these findings could be implemented in clinical practice and hold promise for improving patient outcomes with immunotherapy in the future.

## Figures and Tables

**Figure 1 cancers-16-03955-f001:**
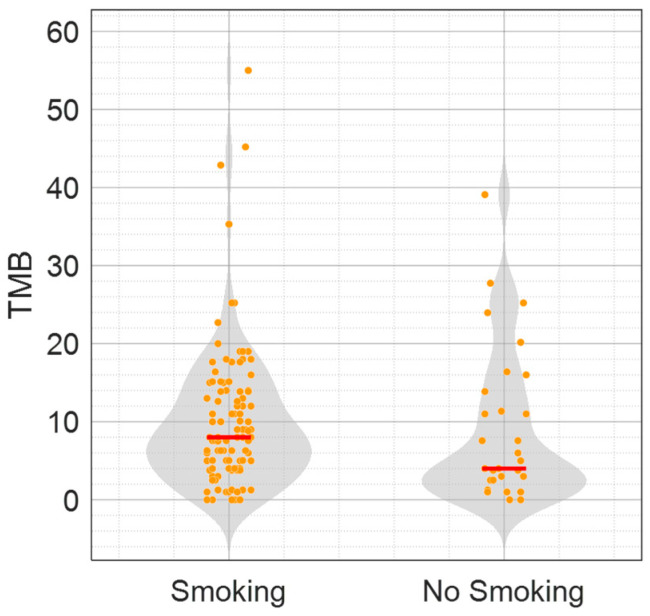
TMB values according to the smoking status population of the real-world aNSCLC patient cohort (142 pts). The violin plots depict the TMB values (Mut/Mb) in S-pts (111) and NS-pts (31). Median values are indicated as red dashes.

**Figure 2 cancers-16-03955-f002:**
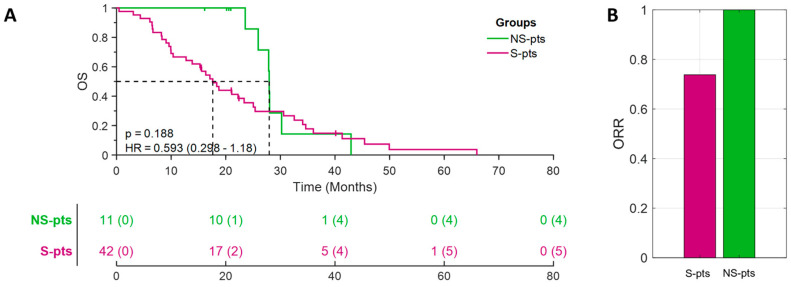
Clinical responses to pembrolizumab in H-TMB patients identified according to smoking status (S- vs. NS-patients): Kaplan-Meier curves for overall survival (OS) (panel **A**) and bar diagram showing ORR values (panel **B**).

**Figure 3 cancers-16-03955-f003:**
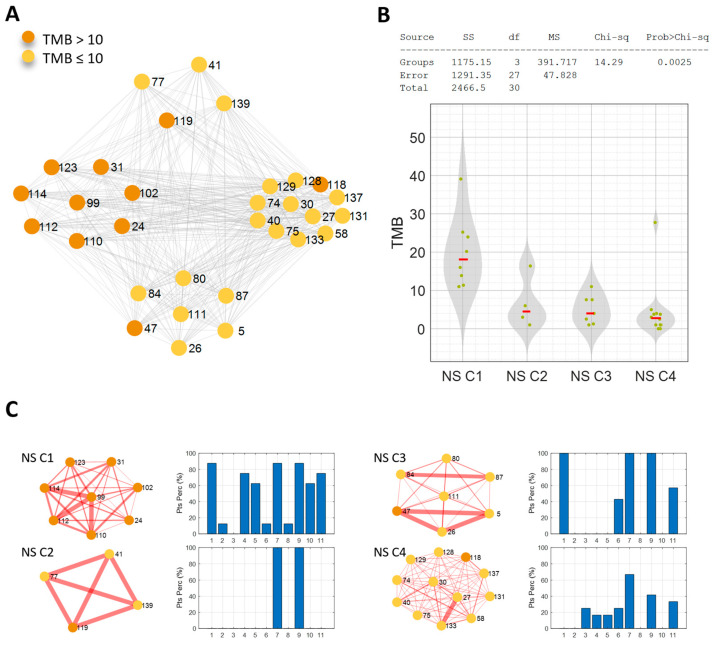
(**A**) Patient similarity network of the never-smoking cohort. Nodes color codes for the TMB level: orange for TMB > 10 Mut/Mb, yellow for TMB ≤ 10 Mut/Mb. (**B**) Kruskal–Wallis test results on the effect of TMB. (**C**) The four identified communities are shown with the associated mutated pathways profile: bar diagrams of the percentage of patients characterized by at least one mutation in each of the analyzed pathways (1-Cell cycle-pathway, 2-Hippo pathway, 3-Myc pathway, 4-Notch pathway, 5-Oxidative stress/Nrf2 pathway, 6-PI3K pathway, 7-RTK/RAS/MAP pathway, 8-TGF beta pathway, 9-p53 pathway, 10-beta-catenin/Wnt pathway, and 11-DDR pathway).

**Figure 4 cancers-16-03955-f004:**
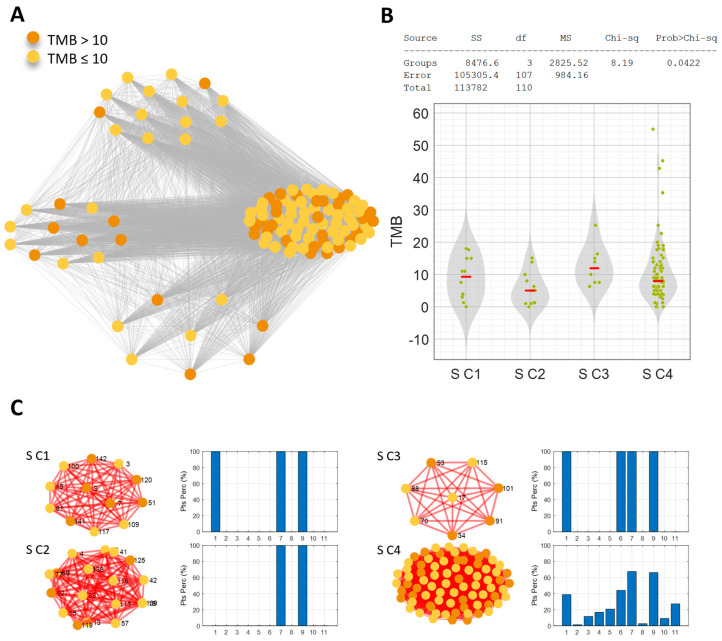
(**A**) Patient similarity network of the smoking cohort. Nodes color codes for the TMB level: orange for TMB > 10 Mut/Mb, yellow for TMB ≤ 10 Mut/Mb. (**B**) Kruskal–Wallis test results on the effect of TMB. (**C**) The four identified communities are shown with the associated mutated pathways profile: bar diagrams of the percentage of patients characterized by at least one mutation in each of the analyzed pathways (1-Cell cycle-pathway, 2-Hippo pathway, 3-Myc pathway, 4-Notch pathway, 5-Oxidative stress/Nrf2 pathway, 6-PI3K pathway, 7-RTK/RAS/MAP pathway, 8-TGF beta pathway, 9-p53 pathway, 10-beta-catenin/Wnt pathway, and 11-DDR pathway).

**Figure 5 cancers-16-03955-f005:**
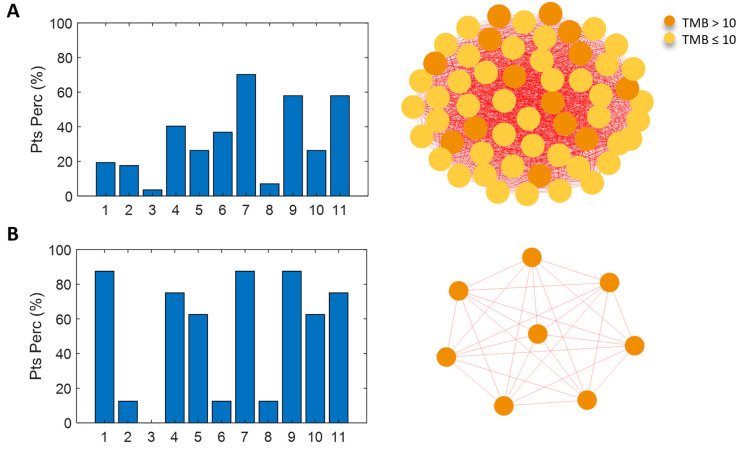
Mutated pathways profile of the OAK/POPLAR NS community (panel **A**) correlated with the mutational profile of the real-world NS C1 (panel **B**): Spearman correlation, ρ = 0.727. The bar diagrams show the percentage of patients characterized by at least one mutation in each of the analyzed pathways (1—Cell cycle-pathway, 2—Hippo pathway, 3—Myc pathway, 4—Notch pathway, 5—Oxidative stress/Nrf2 pathway, 6—PI3K pathway, 7—RTK/RAS/MAP pathway, 8—TGF-β pathway, 9—p53 pathway, 10—β-catenin/Wnt pathway, and 11—DDR pathway). In the network community representation, nodes represent the patients and their color codes for the TMB level: orange for TMB > 10 Mut/Mb, yellow for TMB ≤ 10 Mut/Mb.

**Table 1 cancers-16-03955-t001:** Baseline clinical characteristics of the real-world cohort by smoking status.

Clinical Characteristic	S-pts N = 111	NS-pts N = 31	*p*-Value (S vs. NS)
Age, median (range), years	65 (42–82)	62 (38–84)	0.069
Sex, *n* (%)			
Male	73 (65.8)	15 (48.4)	0.095
Female	38 (34.2)	16 (51.6)	
Histologic profile, *n* (%)			
Squamous	16 (14.4)	3 (9.7)	0.765
Nonsquamous	95 (85.6)	28 (90.3)	
ECOG performance status, *n* (%)			
0–1	103 (92.8)	29 (93.5)	1
≥2	7 (6.2)	2 (6.5)	
Not assessed	1 (1)	0	
TMB, median (range), Mut/Mb	8 (0–55)	4 (0–39.09)	0.129

Abbreviations: S-pts, smoking patient (smokers or former smokers); NS-pts, never-smoking patients; ECOG, Eastern Cooperative Oncology Group; TMB, tumor mutation burden.

**Table 2 cancers-16-03955-t002:** Number (and percentage) of S- and NS- patients characterized by at least one mutation in each of the 11 signaling pathways.

Pathway	S-pts; N = 111 *n* (%)	NS-pts; N = 31 *n* (%)
Cell Cycle	50 (45)	14 (45.2)
Hippo	1 (1)	1 (3.2)
Myc	9 (8.1)	3 (9.7)
Notch	13 (11.7)	8 (25.8)
Oxidative Stress/Nrf2	16 (14.4)	7 (22.6)
PI3K	42 (37.8)	7 (22.6)
RTK/RAS/MAP	86 (77.5)	26 (83.9)
TGF-β	2 (1.8)	1 (3.2)
p53	85 (76.6)	23 (74.2)
β-catenin/Wnt	7 (6.3)	5 (16.1)
DDR *	21 (18.9)	14 (45.2)

* DDR: DNA Damage Response and Repair.

## Data Availability

The data presented in this study are available on request from the corresponding author.

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
