# Peer review of "Exploring Genomic Biomarkers for Pembrolizumab Response: A Real-World Approach and Patient Similarity Network Analysis Reveal DNA Response and Repair Gene Mutations as a Signature"

_cancers, 2024, doi:10.3390/cancers16233955_

Round 1

Reviewer 1 Report

Comments and Suggestions for Authors

This is a well presented and interesting paper, identifying a subgroup of NSCLCs in smoking naïve patients with a distinct mutational profile, high TMB, and relatively good prognosis. While the utility of the discovery of the mutation profile is debatable, given that high TMB is already indicative of IO response, the delineation of the profile has the potential to be informative mechanistically. I think the results could be presented more clearly in places, and the profile they have identified could potentially be investigated/ discussed in more detail, but overall it appears to be a sound piece of work with robust key conclusions. I have split my comments into those I consider more important, and those which are minor or refer to presentation/style.

Main Comments

1. The simple summary contains a lot of specialist terms, and only covers the main findings, not the aims or design. It is also well within the word limit (~85 out of 150) so should be amended accordingly for a non-expert audience.

2. Section 2.4. On what basis were the 11 pathways chosen for analysis? 

3. Presentation of results: There are a few points.

Presentation of the pathway enrichment could be simplified. I understand why the authors have chosen to present just the key results from the H-TMB v L-TMB in the text, and then present all the results from the S v NS groups within the H-TMB cohort in a table. However, this makes the text more difficult to follow, and means that a lot of non significant results are tabulated. Given the small number of significant results, there must be a more elegant way. One possibility would be to show only the significant results from both analyses as Table 2 A and B, or only those with a raw p-value below an arbitrary threshold so that results significant in either analysis are shown in both. This would make it easier for the reader to see what is going on. The full tables can be shown as the supplement. 

Figure 2 Panel B is far too big for the information content. 

Figure 3 Panel B could show individual data points. This would have the major advantage of highlighting that the 8 samples in group C1 are 8 of the top 9 or 10 TMB scores. It is very clear/striking separation. 

Figure 3 Panel C and other figures – the Y axis appears to be a percentage, but the scale is a frequency. This needs to be resolved one way or the other.

Figure 5. I appreciate that it may not look pretty, but ideally, the whole network should be shown, with the two groups of interest highlighted. As specific numbers are irrelevant in the network figures, we don’t need to see them in high resolution (also the panels should be numbered). 

Discussion/further dissection of the C1 profile. From Figure 3 it appears that mutations in Notch, OS/NRF2 and WNT pathways are exclusive to this profile. While the first 2 do not reach significance, they are noteworthy. Are any specific genes being hit repeatedly, can they be rationalised with respect to TMB? Also, within the validation cohort, are there any suggestions as to why H-TMB tumours are at much lower frequency within the subgroup (14/57) compared to 8/8, and how are Notch and OS/NRF2 mutations distributed? I assume the blood origin of the data means it is difficult to compare and will have a lot of samples where very few mutations are identified. If so, this could be flagged to the reader. 

4.     Discussion/Conclusions. As I understand it, TMB data alone would flag the C1 subgroup for IO, without network information, and virtually all C1 tumours have mutations in multiple pathways including p53. While potentially very interesting biologically, is it really likely to be useful in terms of therapeutic decision making? Can this be expanded upon? 

Minor Comments

1. Line 29 (and throughout). PD-L1 > 50% refers to tumour proportion score. This should be made clear at first use in the main text, and ideally removed from the abstract if it can’t be defined. In line 54 it is referred to as expression of 50% or higher, which is also not clear.

2. Line 40. After numbers (11 here), including what percentage of samples this represents would be useful. 

3. Keywords – DDR and NSCLC seem inappropriate – full terms needed. 

4. Line 62 – different PD-L1 expression subgroups are not defined. Clarify?

5. Line 86. . 2. can be removed.

6. Line 111. PD not defined?

7. Line 94 – peculiar is not an ideal choice of words. Novel, unique, distinct? Peculiar suggests counterintuitive, so would need to be explained further.

8. Table 1 – Formatting of 1st row.

9. Line 206 and Figure 1. I assume the TMBs are significantly different between groups. This should be shown. 

10. Line 217/218. What is/are the TMB cut off(s)?

11. Line 238 – enriched FOR DDR…. Also B is missing from in front of Catenin.

12. Line 240/244. I found this difficult to follow as the supplementary figures have no titles/legends. Please add these. Also B- missing again on L243.

13. Section 3.2. and relating to Figure 5. Many readers will not know that Tumour/blood profiles are highly correlated so can be compared. However, they are not the same, and the frequency of mutations identified in the latter is lower, so they will differ in some important respects. This should be clarified in the text and/or figure legend.

14. Line 262 – how does this compare to the other group? If numbers are given for one, an equivalent should be given for the comparator group. 

15. Section 3.3. This would benefit from being cut down where possible, as it is a very long and dense section of text. It could also be split up into paragraphs, e.g. at lines 281/82, 290, 305. 

16. Lines 358-359. As mentioned in the main points, I would argue that characterising the subgroup purely as H-TMB and DDR mutations is an oversimplification, as it is also unique in having high frequencies of WNT, Notch and NRF2 mutations not present in other groups. It can be defined as an H-TMB group, as it contains 10/11 H-TMB samples as far as I can see. However, if you include mutation information in the description, then I think it is important  not to just pick the one with a known functional link to the molecular phenotype.

17. Discussion. As with section 3.3, this could do with being split up. Lines 370, 397? 

18. Lines 402-404. DDR mutations….key events in carcinogenesis. Have NS H-TMB tumours been identified/discussed before? If so, this should be discussed and papers cited. Why highlight in this subgroup specifically? It isn’t clear what point is being made. For the purposes of IO, all but 1 tumour in this group would be flagged as likely responsive. More detailed discussion is perhaps warranted.

19. Lines 411-413. As above. It is not at all clear what mutation status would add. Surely the ~10 patients with DDR mutations and low TMB (~50% of all with DDR mutations?) are of most concern/interest? Again, more discussion of this may be useful. 

Reviewer 2 Report

Comments and Suggestions for Authors

In the current paper, the authors describe their findings regarding the use of TMB and WNT pathway / DDR mutations for the prediction of response to IO in several large clinical trials and 'real world' patients. While TMB has been extensively studied before in these cohorts, a combined analysis is potentially worthwhile. There are several points that should be addressed:

1. While smoking is know to be associated with higher TMB, clinical history taking can be unreliable in that respect. It would be useful to additionally provide DNA smoking signatures from NGS testing.

2. Table 1 does not provide a comparisons between S and NS, please provide p values for each characteristic. 

3. Table 2 provide OS, what about disease-specific survival?

4. Driver mutations are known to be correlated with IO response. Please provide these and compare between cohorts.

Reviewer 3 Report

Comments and Suggestions for Authors

In the manuscript entitled: “Exploring genomic biomarkers for pembrolizumab response: a real-world approach and patient similarity network analysis reveal DNA response and repair gene mutations as a signature.” The authors suggest that DDR genes pathway analysis signature may improve physician confidence in selecting never-smoker patients for immunotherapy.

The manuscript requires further clarification of the analysis conducted to support the conclusions.

Below questions that will help to clarify the approached used:

Page 3 – Line 143:  A total of 11 signalling pathways with frequent genetic alterations were evaluated.

What the authors considered a pathway with frequent genetic alterations? Table S1a reports 73 mutations across 143 cases.  KRAS mutations represent 61% of the mutations (45 out 73) RTK/RAS pathway.  So 28 mutations across 11 pathways. What was the frequency for pathway analysis? There is no DDR genes in Table S1A?

It would help to add a table with the mutation used for pathway analysis. And how authors define germline and somatic (tumour specific mutations)?

There is no report on the number of mutations used for pathway analysis. In Table S2b there is an “n” value but not clear if it is number of patients or number of mutations. Table 1 reports number of cases with mutation in the pathway? Suggesting are not the mutation presented on Table S1A?

Page 5 line 208: “…. 31 patients had no history of tobacco use…”. Was passive or second-hand smoking recorded? Could that explain that 11 NS-pts showed a TMB of higher than 10 Mut/Mb (median TMB: 16.39 Mut/Mb)? If not recorded discuss the potential?

Comparing high and low TMB cases, how they adjust for passenger mutations?

Line 268: This computational approach allowed to build a network based on the molecular similarity of the patients (Figure 3a, b, c).

What was the input for this analysis? What is the computational approach? Details need to be described in the M&M.

Line 279: The fourth community (NS C4, 12 pts) had a weaker profile of 279 mutated pathways.

What a weaker profile means?

Notably, the NS C1 community showed a significantly higher TMB level than the other communities (p-value = 0.0025) (Figure 3b) and most of the patients carried DDR mutations.

What were the DDR mutations? Do these patients have a higher C>T mutations? That could suggest passive smoking.

Line 357: “This study revealed the existence of a subgroup of non-smoking patients characterized by H-TMB and deleterious DDR mutations.”

How sure are the authors that some of DDR mutations are not pathogenic germline variants?  They might want to discuss that possibility as is not clear if they run the panels in tumour only or tumour and matched normal to remove rare germline variants.

Round 2

Reviewer 2 Report

Comments and Suggestions for Authors

I feel the authors have answered my questions to the best of their ability.